# The Role of CT Imaging in Characterization of Small Renal Masses

**DOI:** 10.3390/diagnostics13030334

**Published:** 2023-01-17

**Authors:** Maria Vittoria Bazzocchi, Carlotta Zilioli, Vita Ida Gallone, Claudia Commisso, Lorenzo Bertolotti, Francesco Pagnini, Francesco Ziglioli, Umberto Maestroni, Alberto Aliprandi, Sebastiano Buti, Giuseppe Procopio, Giorgio Ascenti, Chiara Martini, Massimo De Filippo

**Affiliations:** 1Department of Medicine and Surgery, Section of Radiology, University of Parma, Maggiore Hospital, via Gramsci 14, 43126 Parma, Italy; 2Department of Urology, Parma University Hospital, via Gramsci 14, 43126 Parma, Italy; 3Department of Radiology, Istituti Clinici Zucchi, 20900 Monza, Italy; 4Department of Medicine and Surgery, University of Parma–Oncology Unit, University Hospital of Parma, via Gramsci 14, 43126 Parma, Italy; 5Department of Oncology, IRCCS Istituto Nazionale dei Tumori, 20133 Milano, Italy; 6Department of Biomedical Sciences and Morphological and Functional Imaging, University Hospital Messina, via Consolare Valeria 1, 98124 Messina, Italy

**Keywords:** small renal masses (SRMs), renal cell carcinoma (RCC), CT imaging

## Abstract

Small renal masses (SRM) are increasingly detected incidentally during imaging. They vary widely in histology and aggressiveness, and include benign renal tumors and renal cell carcinomas that can be either indolent or aggressive. Imaging plays a key role in the characterization of these small renal masses. While a confident diagnosis can be made in many cases, some renal masses are indeterminate at imaging and can present as diagnostic dilemmas for both the radiologists and the referring clinicians. This review focuses on CT characterization of small renal masses, perhaps helping us understand small renal masses. The following aspects were considered for the review: (a) assessing the presence of fat, (b) assessing the enhancement, (c) differentiating renal tumor subtype, and (d) identifying valuable CT signs.

## 1. Introduction

Renal masses are a frequent finding in radiological investigations and represent a diagnostic challenge for radiologists [1].

Often accidentally detected during an ultrasound (US) or a computed tomography (CT) scan performed for other reasons, renal masses include a vast group of different histological entities, ranging from benign tumors to high-grade malignant tumors [2].

The first includes angiomyolipoma and oncocytoma, the second embodies renal cell carcinoma, with its subtypes clear cell renal carcinoma (RCC), papillary renal cell carcinoma and chromophobe renal cell carcinoma, transitional cell (urothelial) carcinoma, metastasis, and lymphoma.

The true incidence of renal masses is unknown, but growing, especially regarding small renal masses (SMRs), owing to the increasing number of radiological investigations, with cross-sectional imaging, performed in developed countries. The high rate of smoking, hypertension, and obesity, the RCC main risk factors, is also involved [3]. Moreover some syndromes predispose to an increased risk, compared to the general population, for developing kidney lesions. For example, tuberous sclerosis is associated with a high risk of developing AMLs [4] and Von Hippel–Lindau, a rare neurocutaneous inherited syndrome, with malignant tumors such as ccRCC and benign ones such as AMLs [5]. Moreover Birt–Hogg–Dubé, an autosomal dominant condition, characterized clinically by skin fibrofolliculomas, pulmonary cysts, and spontaneous pneumothorax, is associated with different renal cancer, in particular, ccRCC, chRCC, and pRCC [6].

Owing to the small size at the time of diagnosis, almost all patients are asymptomatic; occasionally they present hematuria (micro- or macrohematuria) or abdominal pain [7,8].

In this review, we focus on small renal masses (SRM) that are defined as incidentally CT-detected, contrast-enhancing solid lesions that are ≤4 cm (corresponding to clinical stage T1a) [7].

Among surgically removed SRMs, 70–80% are malignant and 20–30% are benign (if <1 cm, the percentage arises to 45%) [9,10]. Regarding those that are malignant, most are low-grade and unlikely to metastasize. Prognosis is widely favorable, with a survival rate of 95–100% 5 years after diagnosis [7,11].

As a renal mass increases in size, the proportion of malignant versus benign pathologies increases [12]. Frank et al. analyzed 2770 patients undergoing surgery for an SRM and proved the risk of malignancy ranges from 53.8% when the SRM is less than 1.0 cm in diameter up to 80.1% when greater than 3.0 cm and less than 4.0 cm. Concerning metastasis, for every centimeter increase in diameter, the risk of metastasis increases [13].

Despite the increase in diagnostic capacity and in treatment, mortality is stable, suggesting a risk of overdiagnosis and overtreatment [14].

The main management options are active surveillance, surgery, and thermal ablation.

Active surveillance (AS) is the close monitoring of mass size using serial abdominal imaging. Unfortunately, there is not a standardized protocol to decide which patients can benefit from AS, and there are many international trials that establish different eligibility criteria. The American Urological Association (AUA), the National Comprehensive Cancer Network (NCCN), and the American Society of Clinical Oncology (ASCO) guidelines recommend AS for selected asymptomatic patients with renal masses <3–4 cm with a growth rate of less than 0.5 cm/year [12] and for patients who have significant comorbidities and limited life expectancy [15,16].

Surgery is performed via radical nephrectomy (RN) or partial nephrectomy (PN). While RN has served as the definitive management option during the past century, the utilization of PN has been rapidly increasing in recent years [17].

RN is preferred when there is high tumor complexity, no pre-existing CKD or proteinuria, contralateral kidney is normal, and the estimated glomerular filtration rate is likely to be greater than 45 mL/min/1.73 m^2^.

PN should be a priority for the management of cT1a renal masses when intervention is indicated, for patients with an anatomic or functionally solitary kidney, bilateral tumors, known familial RCC, pre-existing CKD, or proteinuria, and in patients who are young, have multifocal masses, or have comorbidities that are likely to affect renal function in the future [18,19].

Thermal ablation is an alternative approach for management of cT1a renal masses smaller than 3 cm [18].

Because of the wide range of choices in management of SRMs, it is essential initially to differentiate between benign and malignant lesions. Biopsy and imaging play a crucial role in this goal.

The purpose of biopsy is to avoid the potential morbidity associated with overtreatment of SRMs [20]. During percutaneous needle biopsy, however, tumor seeding may occur [21]. The mechanism by which this occurs is thought to be the mechanical force of the biopsy directly displacing malignant cells and dissemination of cells caused by bleeding and fluid movement [22].

Precisely because of this risk, very rare but possible, a radiological characterization of SRMs is preferable [23,24].

Not all incidentally detected renal masses demand a complete imaging assessment.

For example, homogenous mass measuring < 20 Hounsfield units (HU) or >70 HU on unenhanced CT is considered benign and does not require further imaging characterization.

Unlike a mass with density > 20 and <70 HU, any heterogeneous mass on unenhanced CT is considered indeterminate and requires a specific multiphase protocol using CT or MRI.

The choice of one or the other is still discussed, but actually CT is more commonly used in radiological departments, owing to its greater availability, lower cost, better spatial resolution, and higher quality images, and showing an average sensibility of 85% [25]. MRI presents a valid alternative for challenging cases [1] and an opportunity in patients with a documented allergy to iodinate contrast or in pregnant women.

### CT-Protocol in SRMs Assessment

There is no general consensus on the CT-protocol to be used. It requires a noncontrast CT scan, followed by intravenous contrast administration. The reference standard for dedicated multiphase renal mass imaging consists of a triphasic CT protocol composed of [26]:A precontrast phase (obtained before the administration of contrast material), to determine the HU of homogenous renal mass or masses containing macroscopic fat (characteristic for angiomyolipoma);A nephrographic phase (100–120 s after the contrast administration), in which there is the maximum and homogeneous enhancement of renal parenchyma, allowing detection of small hypodense masses;A corticomedullary phase (40–70 s), useful to differentiate the RCC subtypes;An excretory phase (7–10 min), recommended in preprocedural planning and in order to distinguish urothelial carcinoma from RCC.

A 2–3 mm section thickness must be used to avoid partial volume artifacts and weight-based dosing of intravenous-iodinated contrast material.

A renal mass is generally considered to be nonenhancing if the change in attenuation is ≤10 HU between the unenhanced phase and the nephrogenic one, and enhancing if the change is >20 HU. If the change is in the middle, the enhancement is considered borderline [2].

## 2. Benign Lesions

Imaging plays a crucial role in characterizing renal masses and guiding management. CT is currently the most commonly used modality for initial diagnosis, and staging, and is considered fundamental to differentiate benign tumors from RCCs and to predict RCC histological subtype and grade when possible [2].

The likelihood of benign histology in small solid renal masses is influenced by size, with a prevalence of up to 40% in lesions less than 1 cm [27,28]. AMLs and oncocytomas comprise most of the benign solid masses, representing, respectively, 44% and 35% [29].

Edges of both benign and malignant lesions are quite the same, in both cases they are sharp, so the radiologist has to look for different morphologic features, such as the presence of a pseudocapsule, which is absent in benign masses because it is typical of ccRCC.

Actually, there is a truly significant morphologic criterion in differential diagnosis, the interface sign, which is widely used in MRI. The interface sign is defined as the morphological relationship between the exophytic renal lesion and the parenchyma surrounding it [30,31].

The interface types can be distinguished according to either angular or round interface shapes. A sharp angle between the intraparenchymal portion of the exophytic renal lesion and the renal parenchyma with a definable apex inside the renal parenchyma exhibits an angular interface, whereas a broad angle reflects round interface [31]. The angular interface describes a triangular shape while the round interface has a wide-rounded shape [32].

The diagnostic challenge is to distinguish RCCs, which may appear in variable forms and mimic other renal tumors, from fat-poor AMLs and oncocytoma [33,34].

Several studies have been carried out based on morphological characteristics of benign and malignant solid lesions. The presence on MRI of an angular interface was reported previously as a sign of benignity [30], but it has also been demonstrated that the angular interface could be used as a reliable benignity sign on US and CT [35]. Kulali et al., similar to many other studies [30,36], emphasized that the angular interface indicates benignity for solid renal masses, and the interface sign may be useful for differential diagnosis of benign and malignant lesions, in addition to other findings from imaging.

### 2.1. Angiomyolipomas

AMLs are benign neoplasms of mesenchymal origin, composed of aberrant blood vessels, mature adipose tissue, and smooth muscle, representing 2% to 6% of all resected tumors in surgical series [37,38].

Typically incidentally found on images, they have female preponderance (1:2, M/F) [39].

They can be distinguished into two distinct epidemiological forms: the sporadic form (80% of cases) and those found in association with genetic syndromes [1]. AML prevalence in patients with tuberous sclerosis varies from 55% to 90% [38]. Tuberous sclerosis is an autosomal-dominant inherited phacomatosis whose clinical hallmarks include adenoma sebaceum, seizures, and mental retardation [40,41]. The tumors found in the individuals with tuberous sclerosis are histologically identical to the sporadic form, but they are more often multiple, bilateral, and present in younger patients. In addition, the AMLs in patients with tuberous sclerosis are often larger at the time of presentation and are likely to grow, so it is not surprising that they are more frequently symptomatic [41,42].

The most specific feature for the diagnosis is considered the detection of fatty tissue (i.e., adypocytes) by CT imaging, although some pathologically proven AMLs do not show fatty tissue on imaging, causing a diagnostic challenge [43]. The detection of macroscopic fat can be confirmed using CT (when the attenuation of the mass is lower than 20 HU) (Figure 1) [1].

The AMLs are usually well defined; calcification or necrosis within the tumor are rare [40], and, as a benign lesion, they are mostly exophytic. Since malignant lesions may also grow exophytic, a useful tip to distinguish AMLs is to look for the presence of an angular interface (Figure 1b). If a hypovascular lesion is endophitic, then it is highly suspicious for a malignant one instead of one that is AML lipid-poor.

The vascular and smooth muscle portions of the tumor variably enhance after administration of contrast material, although less intensely than is seen in RCC or normal renal parenchyma. The vessels in an AML lack a complete elastic layer and tend to be irregular, thick walled, tortuous, and aneurysmal [40,42]. Consequently AMLs are prone to spontaneous bleeding; spontaneous hemorrhage associated with an AML may cause acute flank pain, known as Wunderlich syndrome, a life-threatening complication in larger tumors [44,45].

The diagnostic challenge among these tumors is the fat-poor AMLs lesions that contain less than 25% fat following histological evaluation, accounting for 4–5% of all AMLs, because they currently cannot be reliably distinguished from RCCs [46]. The presence of high attenuation on unenhanced CTs is most suggestive of fat-poor AMLs [47]. Several studies have reported accurate diagnosis of minimal-fat-containing angiomyolipomas using pixel or histogram analyses [48,49], although the findings have not been reproducible [50,51].

### 2.2. Oncocytoma

Oncocytoma, the second most common benign solid renal mass, is composed of oncocytes (polygonal or round cells, with moderate to abundant granular cytoplasm) surrounded by thin capillaries and stroma [42,52,53].

Patients are usually asymptomatic, more frequently men (2:1, M/F), and having a mean age of 65 years at diagnosis [27,54]. Intratumoral hemorrhage and central scars are present in 20% and 33% of all oncocytomas, respectively, and multifocality may occur in 13% of the patients [52].

Oncocytomas are generally small renal masses with a diameter < 4 cm and unifocal, although a small number are multifocal (2–12%) and bilateral (4–12%): several associations have been described among patients with multifocal renal oncocytomas and hereditary syndromes, such as familiar oncocytosis and Birt–Hogg–Dubé syndrome [6,55,56].

In CT, they appear as sharply demarcated lesions with uniform enhancement and often have a central scar (Figure 2).

Differentiation of oncocytoma from RCC is considered difficult. A central stellate scar is a characteristic finding of oncocytoma, which is seen in up to one-third of cases [54]. However, the presence of the scar is considered unreliable to differentiate oncocytoma from RCC: sometimes areas of necrosis occur within RCC and can mimic a central scar, and scars can be present in a small fraction of RCCs [57]. Another tip to distinguish oncocytoma from ccRCC is the presence of a pseudocapsule, which is a malignant feature, even if not always detectable. Therefore, the most common excised benign solid mass is oncocytoma [42].

The main CT feature of oncocytomas is represented by the typical enhancement [1], a hypervascular lesion, particularly the difference between excretory enhancement and attenuation on an unenhanced phase, which has also been described as useful for differential diagnosis with RCCs [58,59].

Some other studies reported that the CT segmental enhancement inversion (SEI) pattern on biphasic MDCT is highly sensitive and specific for the diagnosis of renal oncocytomas smaller than 4 cm [60]; however, other investigators have reported that SEI seems not to be useful for the diagnosis of oncocytoma with CT [61].

Interestingly, in 2020 it was reported that the use of the ratio of lesion to cortex (L/C) attenuation and aorta-lesion attenuation difference (ALAD) on multiphase contrast-enhanced CT helped to distinguish oncocytoma from clear cell RCC in small renal masses [62].

In particular, the study of Gentili et al. demonstrated that the L/C attenuation in the corticomedullary phase seems to be useful for differentiating oncocytoma from clear cell RCC, since oncocytoma appears isodense to normal renal cortex in corticomedullary phase while clear cell RCC appears hypodense [62].

Although oncocytomas are classified as benign tumors [63], case reports have described malignant potential [64]. Similarly, aggressive local behavior may manifest with intravascular extension into branches of the renal vein and invasion of the perinephric fat, the latter occurring in up to 7% of all oncocytomas [27,65].

### 2.3. Inflammatory Conditions and Pseudotumors

A variety of non-neoplastic conditions may mimic solid renal masses [27]. Developmental renal pseudotumors (e.g., prominent columns of Bertin, dromedary humps, persistent fetal lobulations) can be easily differentiated from true renal masses by the delineation of normal renal parenchyma imaging features in the suspicious region; however, other conditions, such as infectious, inflammatory, and granulomatous diseases (e.g., pyelonephritis, abscess, xanthogranulomatous pyelonephritis) may pose a significant diagnostic challenge [27].

It is crucial to contextualize the imaging findings in the appropriate clinical context: focal or multifocal pyelonephritis is usually accompanied by characteristic symptoms, such as chills, fever, flank pain, and pyuria [66].

Focal pyelonephritis can mimic a focal renal mass, especially if confined to a single lobe [67]; CT shows a focal wedge-shaped or rounded area of low attenuation without a well-defined surrounding wall, typically without an overlying bulge on the renal surface, which helps to distinguish it from RCC [67].

It may be difficult to distinguish soft-tissue stranding in the perinephric fat from renal malignancy, such as medullary renal carcinoma [67]. Clinical history should be helpful in establishing the diagnosis of pyelonephritis.

Xanthogranulomatous pyelonephritis, an inflammatory condition usually secondary to chronic obstruction caused by nephrolithiasis, can also present as renal masses in patients with flank pain and fever; is more commonly observed in middle-aged women with urinary stones, infection (most common by Escherichia coli and Proteus), and/or congenital anomalies [68,69].

Generally, this disease is characterized by enlarged kidney, staghorn calculus, destruction of the normal renal architecture, contracted pelvis, and perinephric inflammatory changes [66].

CT evaluation can be considered helpful in the presence of features, such as an abscess replacing the renal parenchyma, with low-attenuation areas (lipid-rich xanthogranulomatous tissue) and calcification in the mass [70,71]. In particular, if calculi are not present, focal XGP with a low-attenuation area in the renal parenchyma may suggest a diagnosis of renal tumor [67].

## 3. Malignant Lesions

Cancer of the kidney and renal pelvis accounts for 3–5% of malignant tumors.

RCC is the most prevalent renal tumor and the World Health Organization divides it into histologic subtypes [63], which include clear cell RCC (ccRCC), papillary RCC (pRCC), and chromophobe RCC (chRCC). Survival is strongly dependent on staging, histologic grade (i.e., Fuhrman grade), presence of sarcomatoid characteristics, and necrosis [72].

The worst prognosis is associated with ccRCC while a better prognosis is associated with pRCC and chRCC. Other less frequent malignant masses include urotelial carcinoma, renal metastases, and renal lymphoma [73].

In general, the probability of malignancy correlates with the diameter of the lesion.

The size of the tumor affects the likelihood of metastatic illness, which is minimal for tumors less than 3 cm. Morphologically, they can appear similar to benign lesions, since they usually have sharp and smooth edges. They frequently present a peritumoral pseudocapsule, which is considered to be a hallmark of malignant renal tumors: according to previous studies, pseudocapsule is present in around 90% of CCRCC, 35%–93% of PRCC, and 30%–53% of ChRCC (Figure 3) [74,75].

Noncontrast CT (NECT) and contrast-enhanced CT (CECT) examination of the lesion may also provide details that aid in the diagnosis [1,27,74,75].

### 3.1. Clear Cell RCC

Approximately 80% of renal tumors measuring 4 cm or smaller are renal cell carcinomas, with the clear cell subtype accounting for the majority of cases (75–80%) [76]; ccRCC originates from the cortex, from the proximal tubule.

Noncontrast CT scans can present cystic components, hemorrhage, and calcifications. A small percentage of RCC can show macroscopic fat. Although the presence of internal macroscopic fat is considered diagnostic of AML, not all fat-containing lesions are AML. Differentiating between a benign AML and fat-containing RCC is often challenging unless there are aggressive features (e.g., irregular or invasive margins, rapid growth, metastatic disease) or there is co-existing calcification within the mass, which is more commonly seen in RCC than AML [77,78,79,80]. Nevertheless, calcifications are rare in AMLs; therefore, if a lesion shows both fat and calcifications, it is more likely a RCC than an AML [81]. Another feature to remember is that a minority of solid clear cell RCC may measure between 10 and −20 HU, meaning that not all water attenuation lesions are renal cysts/cystic lesions [82].

When a lesion measures between 10 and 20 HU, it usually has a thin undetectable wall, and if consistently homogenous, the diagnosis of simple renal cyst may be made reliably. Despite this, some RCCs presenting these features are reported in literature [82,83]. The problem in diagnosis between a simple cyst and a malignant mass could be solved with CECT.

Moreover, in NECT, it could be challenging to discriminate between a hemorrhagic cyst and some RCC. Recently, Youmada et al. proved that when a homogenous renal mass measures >70 HU at NECT, this amount of attenuation is indicative of a hemorrhagic cyst [84].

In contrast-enhanced CT, ccRCC enhances vividly, with the maximum of intensity in the corticomedullary phase (with an interval between 116.5 and 133.5 HU, according to Young et al. [85]). It remains enhanced in the portal–venous phase and progressively decreases in the excretory phase [86] (Figure 4).

This attenuation trend is the same for oncocytoma, while pRCC and chRCC show a lower attenuation in each phase.

Chromophobe and papillary lesions both reach their highest enhancement during the nephrographic phase, as opposed to ccRCC, which reaches its maximal enhancement during the corticomedullary phase [85]. However, since renal oncocytoma (RO) shares the same attenuation pattern as ccRCC, but they have a significantly different prognosis, the goal of the radiologist is to distinguish one from the other. Some authors have validated new methods, CT based, to make a confident diagnosis based on attenuation threshold.

As absolute attenuation measurement might be affected by patient and lesion parameters (i.e., cardiac function, weight, state of hydration, and renal function), it is not a valid criterion for distinguishing ccRCC from RO.

Gentili et al. proposed to normalize the attenuation, using the difference between the enhancement of the aorta compared with the enhancement of the lesion (aorta lesion attenuation difference, ALAD) and the ratio of lesion to normal cortex attenuation (L/C) [62]. Their study proved that in corticomedullary phase, RO appeared almost isodense to normal renal cortex (L/C = 0.92 ± 0.12), while clear cell RCC appeared hypodense (ratio L/C 0.69 ± 0.20). This phenomenon could be explained because ccRCC has some areas of micronecrosis while RO has a normal vascular structure, similar to normal renal parenchyma [87,88]. On the contrary, in the nephrographic phase, RO showed an L/C ratio lower compared to ccRCC, which had almost the same corticomedullary phase L/C ratio. This may be explained by the fact that oncocytoma has normal vessels compared to ccRCC. These preliminary data have to be validated by multicentric studies with a larger number of cases.

Thus, in general, renal oncocytoma showed an early washout threshold, while ccRCC a prolonged enhancement [89].

### 3.2. Papillary RCC

Papillary RCC is the second most frequent RCC subtype (17–18% of RCCs [90,91]). pRCC originates from proximal tubule and includes types I and II. Type I (basophilic) is associated with a better prognosis, usually it is sporadic but could be associated with MEN syndrome in small cases. Type II (eosinophilc) is less common and, on the contrary, has a worse prognosis [92]. These two types cannot be differentiated on imaging.

pRCC is usually hypovascular and in noncontrast CT it may present with hyperdense foci (measuring from 45 to 90 HU), which are circumscribed hemorrhagic spots. Complicated cysts could show hyperdense foci as well, but at this point contrast enhancement becomes crucial. pRCC presents progressive enhancement and reaches the maximum attenuation in nephrographic phase [47] (Figure 5), while cysts do not enhance.

The mean enhancement of ccRCC is significantly greater compared to pRCC in every phase [85].

J. Ruppert-Kohlmay et al. found that tumor attenuation smaller than 100 HU in the corticomedullary phase, when normalized for aortic enhancement, was 95.7% specific for papillary RCCs, compared to ccRCC [93].

Since they are hypoenhancing, they may have an unclear attenuation range (10–20 HU) or no attenuation change (10 HU), so a contrast-enhanced ultrasound or MRI may be required to confirm the diagnosis [94].

### 3.3. Chromophobe RCC

ChrRCCs represent 4–8% of RCCs [90,91] and originate from collecting-duct intercalated cells. It is the RCC subtype associated with the best prognosis, a 5-year survival rate around 90% [95,96]. ChRCC can be seen in patients with Birt–Hogg–Dubé syndrome. These patients have small papular skin lesions called fibrofolliculomas—lung cysts with spontaneous pneumothorax.

ChrRCC shows more commonly homogeneous–moderate enhancement, lower compared to ccRCC, with the attenuation peak in the nephrographic phase. Based on the enhancement threshold, it is easily differentiated from ccRCC [97,98].

ChRCC at the time of diagnosis generally has a diameter >4 cm and may present calcifications and could occasionally contain a central scar. Typically, it has a homogeneous enhancement and a homogeneous cut surface without hemorrhage or necrosis [99] (Figure 6 and Figure 7).

CT imaging plays a pivotal role in the diagnosis of chRCC since its histologic appearance has some variations, such as eosinophilic variant, which mimics renal oncocytoma. In some cases, both chRCC and oncocytoma share even the same markers, so the radiologist has an essential position in defining the lesion [100,101].

### 3.4. Urothelial Renal Carcinoma

Transitional cell carcinoma (TCC), also known as urothelial cell carcinoma (UCC), arises from the epithelial cells lining the urinary tract. It is a rare tumor (less than 5% of renal tumors) and usually arises from the renal pelvis as an intraluminal mass of the renal-collecting system.

TCC is more frequently low grade, but in 15% of cases it is aggressive with retroperitoneal invasion. Commonly, it is multifocal with a high risk of recurrence. Synchronous bilateral TCC has been reported to occur in 1–2% of cases of renal lesions and 2–9% of cases of ureteral lesions; 11% to 13% of patients with upper-tract TCC subsequently develop metachronous upper-tract tumors [102].

In noncontrast CT, it is usually hyperattenuating (5–30 HU) to urine and renal parenchyma, but less attenuating than other pelvic-filling defects such as clot (40–80 HU) or calculus (100 HU) [103]. In 2% of cases, it could be calcified; thus, it can be difficult to distinguish it from calculi [102,104].

The urographic phase is crucial to diagnose TCC: it is a typical filling defect that can expand and compress renal sinus fat. It may present as focal or diffuse mural thickening or as focally obstructed calices. Both TCC and ccRCC can show early enhancement and early washout, but normally RCC reaches higher HU values [105].

Since, even in small TCC, hydronephrosis is the most frequent finding and hydroureter can often be seen to the point of obstruction, this feature can be useful to differentiate TCC from RCC. A small renal RCC tumor is quite unlikely to cause hydronephrosis [106].

### 3.5. Lymphoma

The genitourinary system is commonly affected by extranodal spread of lymphoma in which the kidneys are the most commonly involved organ [107]. Primary renal lymphoma, that is, the involvement of the kidney without systemic disease, is rare (1% of cases of extranodal lymphoma). Secondary lymphoma is much more common and originates from the direct spread of retroperitoneal adenopathy or hematogenous spread from systemic disease [108].

Often lymphoma is asymptomatic; however, patients may present with flank pain, hematuria, night sweats, and fever [109]. Frequently, lymphoma is accidentally detected during radiological examinations.

CT is the first imaging modality for the assessment of patients with suspected lymphoma. A late arterial phase, after the administration of intravenous contrast is essential for the detection of subtle lesions and to evaluate the vasculature differentiating lymphoma from hypervascular primary renal tumors [108]. Renal lymphoma is typically a hypovascular tumor. This feature can help to differentiate it from the more common hypervascular tumors such as RCC, oncocytoma, and angiomyolipoma.

Imaging findings are unspecific, even if six major patterns have been described: multiple lesions, solitary lesion, direct extension from retroperitoneal adenopathy, perinephric disease, nephromegaly, and renal sinus involvement. The most common imaging pattern that occurs in this kind of tumor is multiple solid parenchymal masses (50–60% of cases) [110], more often bilateral but also monolateral. On unenhanced CT, lesions typically are homogeneous and have higher attenuation than the surrounding renal parenchyma. Of course, it is necessary to make differential diagnosis with multifocal renal lesions, first with metastatic disease; in 15–20%, lymphoma presents as a solitary mass.

### 3.6. Renal Metastases

Primary malignancies that most commonly metastasize to the kidney reflect the pattern of cancer occurrence in the general population, with lung, breast, gastrointestinal tumors, and melanoma [42,111,112].

Commonly, the primary tumor is already known or diagnosed at the same time as the renal lesion, with more than half of the cases occurring in patients older than 60 years [42,113]. Bilateral or multiple masses are found in 23% and 30% of the patients, respectively [113].

They are typically small, multifocal, and bilateral. If they manifest as a single lesion, then it is hard to differentiate from RCC [27,42].

Renal metastases occur more commonly at the junction of the renal cortex and medulla, often showing ill-defined borders and low-level enhancement, except in the case of hypervascular primary tumors (e.g., RCC, thyroid, choriocarcinoma) [42,114].

These features may help to suggest the diagnosis and differ from the most common well-defined appearance of cortical-based RCCs, although a definitive diagnosis may require a biopsy [27].

## 4. Discussion

A portion of the incidentally detected, small, solid renal masses has overlapping imaging features between benign and malignant tumors. Despite advances in CT, they remain indeterminate [29].

The aim of our study is to give the reader a useful framework to assist in correct diagnosis with CT imaging, and to reduce the number of renal biopsies leading to the possibility of seeding.

Since morphologic appearance of renal masses is not helpful in differentiating benign from malignant lesions, as they share the same sharp edges, it is crucial to discern between hypovascular or hypervascular solid lesions.

Hypovascular lesions are strongly related with malignant potential; in some cases, biopsy is not even necessary to recognize them.

ChRCC and pRCC, and also ccRCC in some cases, are considered hypovascular since they show delayed and prolonged enhancement with a peak in nephrographic phase, occasionally in the excretory phase (pRCC poorly vascularized).

Among hypovascular lesions, some help for the radiologist can come from morphologic features. ChRCC usually presents as a large lesion and can show macroscopic calcifications, while pRCC is mostly a small renal mass that may show hyperdense spots (from 45 to 90 HU) in noncontrast CT. The problem in diagnosis between hemorrhagic cyst and pRCC could be solved with CECT.

Macroscopic (extracellular) fat in the mass is highly typical of AMLs. At this point, the first difficulty may occur since there are some exceptions: firstly, angiomyolipomas free of macroscopic fat may occur, even if rarely. Lipid-poor AMLs have the same enhancement pattern of malignant hypovascular lesions, causing one of the principal diagnostic challenges. A useful tip is to evaluate the morphologic features of the mass: if the mass is endophytic, is highly suggestive of ccRCC instead of lipid-poor AML, while if the mass is exophytic and the angular interface appears, the radiologist can undoubtedly consider lipid-poor AML instead of malignant lesions.

The presence of an angular interface, indeed, is reported as a sign of benignity, which could be detected using CT, and is typical of lipid-poor AMLs.

On the other hand, the presence of fat could be found among malignant solid masses, generally associated with calcifications due to degeneration within the mass. The presence of both calcification and fat tissue should lead the radiologist to RCC.

When considering hypervascular lesions, both ccRCC and oncocytoma show the same contrast-enhancement pattern.

As already mentioned, an interesting method to help distinguish between the two could come from the ratio of lesion to cortex (L/C) attenuation and aorta/lesion attenuation difference (ALAD) on multiphase CECT. In nephrografic phase renal oncocytoma showed an early washout threshold, while ccRCC produced a prolonged enhancement, although this method proposed by Gentili et al. is still under validation in a multicentric study. At this point, a useful tip for the radiologist is once again the morphologic features: the presence of a pseudocapsule is peculiar to ccRCC. Moreover, if lesions are multifocal and hypervascular, then they are highly suspicious for malignant ccRCCs instead of oncocytomas.

Another strategy to remember is that oncocytomas rarely reach a diameter > 4 cm, while ccRCC could reach larger dimensions.

CT is also the cornerstone for differential diagnosis of oncocytoma and chRCC, which may show histologically the same traits, characterized by eosinophilic cells. This is a typical circumstance of when biopsy could not be diriment, but instead CT imaging is. In fact, they show opposite contrast enhancement patterns: oncocytoma is hypervascular and chRCC is hypovascular.

Throughout the last few decades, biopsy and histology have been the leading techniques for a correct diagnosis, despite their complications, mostly the risk of seeding, and the possibility of inaccuracy in a few cases, in particular for the differential diagnosis between oncocytoma and chRCC, both sharing eosinophilic cells, as previously noted.

During the last few years, the evolution of the CT imaging technique has strengthened the noninvasive assessment of SMRs and management options. A large number of studies have been conducted on the trend of this topic, proposing some interesting features such as the ALAD ratio or L/C ratio, which could be promising but still need to be validated.

Despite all this, a small percentage of SRMs still remain uncharacterized, requiring invasive methods together with their complications.

Figure 8 summarizes the principal steps that the radiologist should follow to understand small renal masses.

Relatively recent imaging modalities that have been proposed to differentiate malign from benign renal masses are represented by ^99m^Tc-sestamibi SPECT/CT and radiomics.

The first one has been described in 2015 [115] and relies on the high concentrations of mitochondria in renal oncocytomas that avidly accumulate ^99m^Tc-sestamibi and appear “hot” on SPECT, while most RCCs are relatively devoid of mitochondria [116]. In this study [115] ^99m^Tc-sestamibi SPECT/CT showed a sensitivity of 87.5% and specificity of 95.2% for diagnosing renal oncocytomas and hybrid oncocytic/chromophobe tumours (HOCTs), however there are some limitations: assessment of renal masses smaller than 1.5-2 cm and endophytic masses is quite difficult due to the intrinsic spatial resolution images of SPECT and normal high background renal uptake of radiotracer [117].

Opposed to SPECT/TC, that is mostly based on a qualitative approach, as there are no strict cut-off values, radiomics is totally dependent on quantitative data extrapolation from images.

Radiomics allows a new approach to develop predictive tools by correlating imaging features to tumor histology. In particular, pixel distribution and pattern-based texture analysis appear as quantitative methods for the detection of tissue differences that cannot be find out by visual assessments [118].

A recent meta-analysis determined that studies investigating the differentiation of benign versus malign renal tumors showed promising results overall and that radiomics might bring an added value in combination with human assessment [119]. Nevertheless in literature there is a huge inhomogeneity in terms of the methodology of the used image reconstruction, feature extraction, the algorithms used and about the imaging protocols and the deriving phases from which the features were extracted, leading to a difficult standardization of results and low reproducibility. Still, the low cost, non-invasiveness and large applicability (not only to histology but also to tumor grade, genetic patterns and molecular phenotypes as well as clinical outcomes) guarantee to radiomics a role of primary importance in future diagnostic approach to small renal masses.

## 5. Conclusions

The best approach for CT characterization of small renal lesions should imply some relevant acknowledgments.

Sharp edges in renal masses is not specific for benignity; in fact, the vast majority of RCCs show this feature. On the contrary, infiltration of perinephric fat (T3), Gerota’s fascia, nearby organs (adrenal glands, pancreas, liver, etc.) are indisputable marks of malignity (T4).

The presence of macroscopic fat is peculiar to angiomyolipoma.

Microscopic fat, on the other hand, may be present both in lipid-poor angiomyolipomas and in some clear cell RCCs, but just pathology could point it out. In fact, microscopic fat is only occasionally proven by CT, measuring HU, or RM (using GE T1w out-of-phase sequences).

Therefore, lipid-poor angiomyolipomas present a challenge for noninvasive diagnostic imaging; hence, some tricks may guide the diagnosis. If the lesion exhibits an angular interface, which is more easily detectable by MRI, the diagnosis of lipid-poor angiomyolipomas can be supported; if the lesion shows a hypovascular or hypervascular pattern after CECT with some calcifications, then RCC is a highly probable diagnosis.

Solid lesions showing a delayed enhancement pattern are usually low-grade RCCs, as are chRCC, pRCC, and also ccRCC; less frequently, hypovascular lesions are benign (angiomyolipoma or xanthogranulomatous pyelonephritis).

Currently, histology, after imaging-guided biopsy, is crucial to discriminate between oncocytoma and hypervascular ccRCC; thus, another future challenge for radiologists will be to distinguish one from the other by only relying on CT imaging.

## Figures and Tables

**Figure 1 diagnostics-13-00334-f001:**
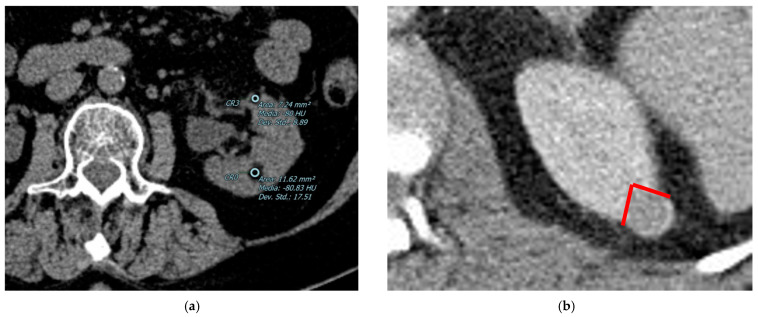
Abdominal CECT of a 77-year-old female patient: (**a**) the left kidney contains two cortical formations, the first in the middle-upper third (12 mm) and the second in the middle-lower third (11 mm), both with a preponderant macroscopic fat component. Both lesions show an average density lower than 20 units (−80 and −80 HU, respectively) and hence consistent with angiomyolipomas. (**b**) An example of a typical lipid-poor AML: the lesion is exophytic with a peculiar angular interface (red lines). Both these features help the radiologist in the diagnosis.

**Figure 2 diagnostics-13-00334-f002:**
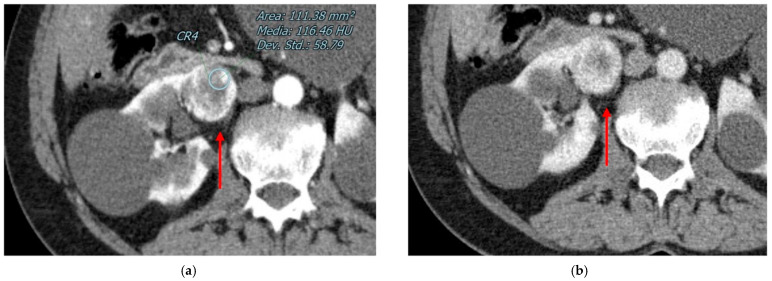
Abdominal CECT of 66 year-old male patient with a 16 mm nodular lesion in his right kidney. After the administration of contrast, the small lesion enhanced homogeneously, showing the peculiar hypervascular pattern (**a**) and revealing a central scar (**b**). These features are suitable for oncocytoma.

**Figure 3 diagnostics-13-00334-f003:**
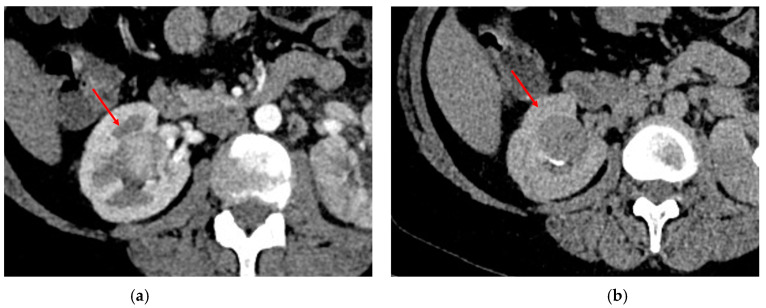
Example of endophytic malignant renal cell tumor with pseudocapsule (red arrow) at CECT in nephrographic phase (**a**) and urographic phase (**b**).

**Figure 4 diagnostics-13-00334-f004:**
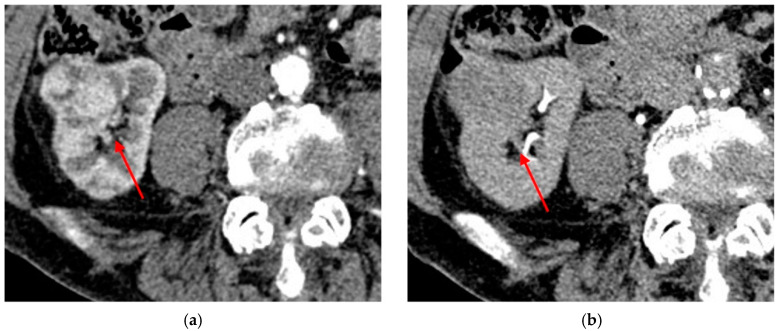
Abdominal CT of ccRCC: (**a**) in the corticomedullary phase, the lesion demonstrates an intense contrast enhancement; (**b**) in the excretory phase, the contrast enhancement progressively decreases.

**Figure 5 diagnostics-13-00334-f005:**
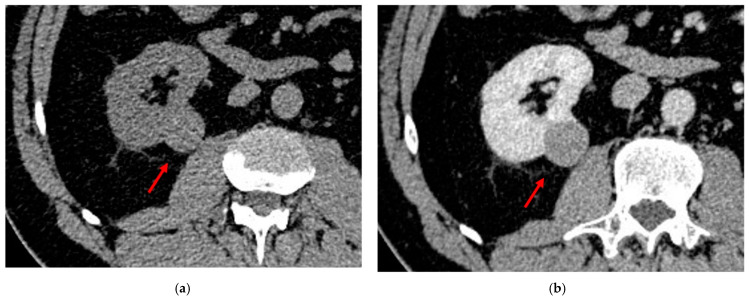
Abdominal CECT of pRCC demonstrate a cortical small renal lesion. (**a**) In a pre-contrastographic phase, the formation is isodense with the adjacent renal parenchyma. (**b**) The same lesion progressively enhances homogeneously, but due to the hypovascular nature, shows overall hypoenhancement compared to the adjacent normal renal cortex.

**Figure 6 diagnostics-13-00334-f006:**
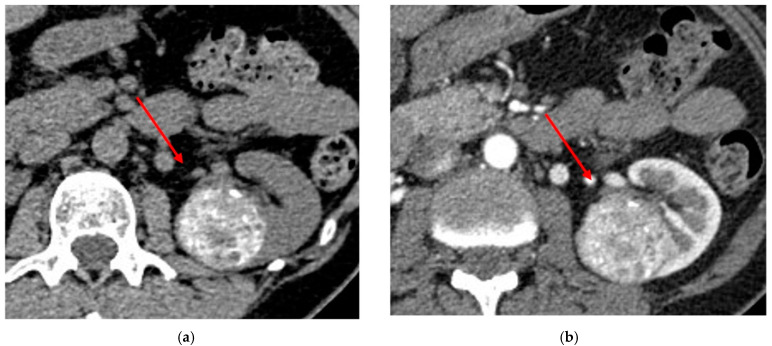
Abdominal CT of an incidentally detected intraparenchymal renal lesion: (**a**) shows multiple macroscopic calcifications in a noncontrast CT; (**b**) shows that the mass is hypovascular in the arterial phase. The lesion is histologically confirmed as chRCC.

**Figure 7 diagnostics-13-00334-f007:**
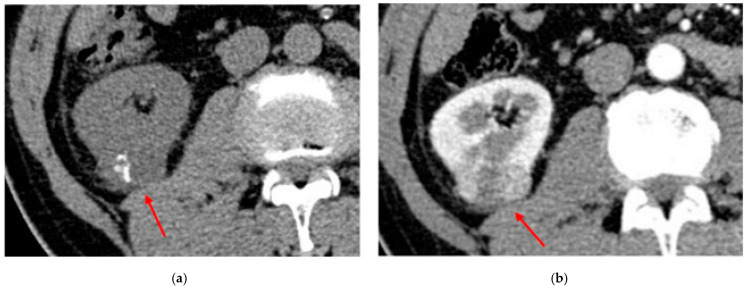
Another example of histologically confirmed chRCC showing (**a**) multiple macroscopic calcifications in a noncontrast CT and(**b**) the typical enhancement pattern.

**Figure 8 diagnostics-13-00334-f008:**
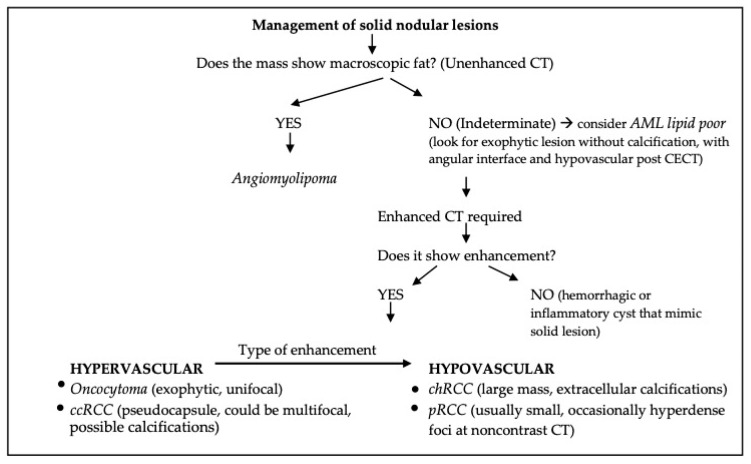
Modified Nicolau et al. algorithm for assessment of SRMs [1].

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
