# Peer review of "The Role of CT Imaging in Characterization of Small Renal Masses"

_diagnostics, 2023, doi:10.3390/diagnostics13030334_

Round 1

Reviewer 1 Report

This is a very good summary concerning the role of CT in the characterisation of renal tumours. I am missing (mainly in the Discusion) some info about Radiomics as well as the use of 99mTc Sestamibi SPECT/CT in this specific field of oncological imaging. 

Author Response

Dear Reviewer, 
the response to your comment can be found in the attachment. 

Thank you 

Reviewer 2 Report

The novelty of this review was a moderate. There have been several reviews in this field. Authors should focus on the diagnostic performance of CT imaging in SRM.

Each part of renal diseases should be with CT imagings. 

Differential diagnosis of listed renal diseases should be summarised in a table.

Author Response

(The authors gave the same response as above.)

Reviewer 3 Report

1.Small renal masses are increasingly detected incidentally at imaging. They vary widely in histology and aggressiveness, and include benign renal tumors and renal cell carcinomas that can be either indolent or aggressive. Imaging plays a key role in the characterization of these small renal masses. While a confident diagnosis can be made in many cases, some renal masses are indeterminate at imaging and can present as diagnostic dilemmas for both the radiologists and the referring clinicians.This review focus on  CT charaterization of  small renal masses,It may help us  understanding small renal mass.The following  aspects were consider for the review:a.assessing the presence fat;b.assessing the enhancement;c.differntiating renal tumor subtype;d. valuable CT sign.

2.Figure 2, the arrow is missing in the picture, I don't think that  there is no sign of a central scar

3.It might be beneficial to include arrows in Figures

Author Response

(The authors gave the same response as above.)

Round 2

Reviewer 2 Report

This version is better. There are several typos such as 'mascroscopic' in Figure 7.